# Assessing Two Different Aerial Toxin Treatments for the Management of Invasive Rats

**DOI:** 10.3390/ani12030309

**Published:** 2022-01-27

**Authors:** Tess D. R. O’Malley, Margaret C. Stanley, James C. Russell

**Affiliations:** 1School of Biological Sciences, University of Auckland, Waipapa Taumata Rau, Private Bag 92019, Auckland 1142, New Zealand; mc.stanley@auckland.ac.nz (M.C.S.); j.russell@auckland.ac.nz (J.C.R.); 2Department of Statistics, University of Auckland, Waipapa Taumata Rau, Private Bag 92019, Auckland 1142, New Zealand

**Keywords:** density, eradication, invasive species, predator-free, *Rattus rattus*, rodent, sodium fluoroacetate (1080), spatially explicit capture–recapture

## Abstract

**Simple Summary:**

In Aotearoa–New Zealand, the helicopter application of the toxin sodium fluoroacetate (1080) is a common method for controlling invasive mammals. However, the application of 1080 using current methods leaves some surviving mammals, meaning eradication cannot be achieved. A new application method, called 1080-to-zero, aims to eradicate target mammals or reduce them to near-zero levels. This study monitored the response of invasive black rats (*Rattus rattus*) to a 1080-to-zero application and a standard 1080 application. In this case it found that the 1080-to-zero method did not improve rat removal compared to the standard application, and did not reduce rats to near-zero levels. However, these results differ from a 1080-to-zero application in another part of the country, which did achieve near-zero abundance for rats. Questions remain about how local factors affect this tool, and how it can be further improved.

**Abstract:**

Aotearoa–New Zealand has embarked on an ambitious goal: to completely eradicate key invasive mammals by 2050. This will require novel tools capable of eliminating pests on a large scale. In New Zealand, large-scale pest suppression is typically carried out using aerial application of the toxin sodium fluoroacetate (1080). However, as currently applied, this tool does not remove all individuals. A novel application method, dubbed ‘1080-to-zero’, aims to change this and reduce the abundances of target pests to zero or near-zero. One such target is black rats (*Rattus rattus*), an invasive species challenging to control using ground-based methods. This study monitored and compared the response of black rats to a 1080-to-zero operation and a standard suppression 1080 operation. No difference in the efficacy of rat removal was found between the two treatments. The 1080-to-zero operation did not achieve its goal of rat elimination or reduction to near-zero levels, with an estimated 1540 rats surviving across the 2200 ha treatment area. However, 1080 operations can produce variable responses, and the results observed here differ from the only other reported 1080-to-zero operation. We encourage further research into this tool, including how factors such as ecosystem type, mast fruiting and operational timing influence success.

## 1. Introduction

Invasive mammals present a global threat to biodiversity, especially in island ecosystems such as those of Aotearoa–New Zealand [1]. Native New Zealand species are particularly vulnerable, as they evolved in the absence of terrestrial non-volant mammals [2]. Predation and herbivory from introduced mammals has been a leading cause of extinctions and habitat degradation [3,4]. In response, New Zealand has made significant developments in invasive mammal control, pioneering techniques such as eradication from islands [5,6] and fenced mammal-free sanctuaries [7]. Recently, New Zealand has begun pursuing a new era of management: the eradication of invasive mammals from large, contiguous landscapes.

In 2016, the government announced Predator Free 2050 (PF2050): an initiative to eradicate rats, possums and mustelids from all of New Zealand by 2050. This encompasses seven species: black rat (*Rattus rattus*), brown rat (*R. norvegicus*), Polynesian rat (*R. exulans*), brushtail possum (*Trichosurus vulpecula*), ferret (*Mustela furo*), the least weasel (*M. nivalis*) and stoat (*M. erminea*). This has received strong support but represents a sharp departure from historic management practices [8]. Typically, eradication has been employed in small, isolated areas where the chances of success are relatively high and reinvasion relatively low. Larger landscapes have received suppression management: the control of pests to low but non-zero abundances [9,10]. At 11.5 and 15 million hectares each, New Zealand’s two ‘mainland’ islands are far from small, and the management areas within them are not isolated. The complex, contiguous landscape makes it challenging to meet two of the key requirements for eradication: the total removal of individuals within the treatment area and the preclusion of reinvasion [11,12].

In response to these challenges, many of the emerging mainland eradication projects have adopted what is termed a ‘remove and protect’ management strategy [13]. This strategy aims to remove resident pests and protect against reinvasion by detecting and removing invaders as they arrive. Using this strategy, pest elimination zones can be established and expanded, ultimately enabling mainland eradication [13]. The remove and protect strategy is best served by treating large areas, ideally in the tens of thousands of hectares. This reduces reinvasion pressure by shrinking the boundary-to-treatment-area ratio and improving the flexibility to use natural barriers to movement [14,15].

As treatment area increases, financial cost becomes a limiting factor. Although ground-based traps and bait stations can be effective at eradicating mammals [6], they are labour-intensive, especially when targeting pests with small home ranges [13]. Rats have the smallest home ranges of the PF2050 target species (typically < 1 ha), and their elimination can require device densities of at least one device every 50 m [6,16]. In areas without access to ample volunteers, these labour costs are typically prohibitive over a large scale, even when considering the multi-trap devices currently on the market [17]. Additionally, parts of the New Zealand landscape (e.g., cliffs) make ground-based access extremely challenging.

In contrast, the aerial application of toxins is comparatively cost-effective [13,18,19]. However, there are critical gaps in what this tool can achieve. In New Zealand, the only approved toxins for aerial application are brodifacoum and sodium fluoroacetate (also known as 1080). Brodifacoum can be reliably used to eradicate mammals from large islands (e.g., 11,300 ha Campbell Island [20]), but its use is restricted on the mainland due to its long environmental persistence [9]. In comparison, 1080 has a low environmental persistence and is regularly used for the mainland suppression of possums, rats and stoats [10]. However, it tends to leave small surviving populations, making eradication unachievable [9].

Motivated by PF2050, attention has turned to developing 1080 application methods capable of achieving results more typical of brodifacoum [9,21]. The success of aerial toxin applications is known to be highly dependent on the precise operational methods followed [22], and the past few decades have seen significant improvements in the safety, reliability and efficacy of 1080 applications [18]. Currently, owing to their different toxicology and purposes, brodifacoum and 1080 application methods differ, with the former often including features such as two toxic applications and higher bait sow rates [23]. A novel eradication-focused 1080 application may be able to provide a cost-effective, large-scale elimination tool for mainland eradications.

Zero Invasive Predators, a research NGO, has developed a new 1080 application method which borrows heavily from brodifacoum applications. Dubbed ‘1080-to-zero’, this method features several improvements: pre-feeding with non-toxic bait twice instead of once; overlapping helicopter flight paths to reduce gaps in bait distribution; and applying more bait per hectare (in contrast with Dilks [24]). If post-treatment monitoring indicates the presence of survivors, the pre-feed and toxic bait are re-applied several weeks later, producing a ‘double 1080′ operation [21].

In 2019, two large-scale 1080-to-zero operations were carried out: one in the Perth River Valley, on the West Coast of the South Island, and one in the Kaitake Ranges of Taranaki, in the North Island. To critically evaluate the success of a 1080-to-zero operation, and indeed any 1080 operation, it is necessary to know the density of pests prior to the 1080 application, the extent and duration of the reduction to zero density, and the rate at which the population recovers without ongoing management. The 2019 South Island short-term results for possums, black rats and stoats have been published elsewhere [21]. Here, we evaluate the short- and long-term effectiveness of the 2019 North Island 1080-to-zero application on black rats, one of the PF2050 target species most difficult to eradicate using ground-based tools, and thus most in need of aerial alternatives [25].

## 2. Materials and Methods

### 2.1. Study Sites

The study took place in Te Papakura o Taranaki, a national park on the west coast of the North Island of New Zealand (Figure 1). The 34,000 ha park is centred on Taranaki Mounga, a large dormant volcano hereafter referred to as the Mounga. To the north are the 2200 ha Kaitake Ranges, included in the national park. One pest monitoring site was selected in the Kaitake, and one at the base of the Mounga. These were matched by approximate elevation (330 m at Kaitake vs. 615 m at Mounga), distance from bush edge (837 m vs. 1260 m respectively) and habitat type. Both sites were in mixed podocarp hardwood forests, but the fine-scale vegetation composition varied [26]. The Kaitake site was situated in a tawa (*Beilschmiedia tawa*)-dominated lowland forest, with rewarewa (*Knightia excelsa*), pukatea (*Laurelia novae-zelandiae*) and hinau (*Elaeocarpus dentatus*) being common [26]. The Mounga site was located in a lower montane forest, dominated by kahikatea (*Dacrycarpus dacrydioides*), rimu (*Dacrydium cupressinum*) and kamahi (*Pterophylla racemose*) [26]. In March and April of 2019, unusually heavy fruit production was observed at both sites. This is consistent with Department of Conservation seed sampling, which indicated 2019 was a heavy mast year [27]. Mass synchronised seed and fruit production was observed nationwide, including in beach and rimu forests and tussock grasslands [27].

### 2.2. 1080 Aerial Application Treatments

In 2019, as part of a PF2050 possum elimination project called Restore Kaitake, the Kaitake Ranges were treated with a 1080-to-zero aerial toxin application. Although primarily targeting possums, this application used the same methods as 1080-to-zero operations targeting possums, rats and stoats [21]. Non-toxic pre-feed was applied on 4 and 17 April 2019, followed by toxic bait on 1 May 2019. Subsequent possum monitoring by the Restore Kaitake project identified numerous survivors: 21 days post-toxin application monitoring detected possums at 27 of 60 trail cameras, with one camera per 42 ha throughout the Kaitake Ranges (personal communication [28]). This level of survivorship triggered a second application, with pre-feed applied on 28 June and 18 September, and toxic on 31 October. Bait sow rates were reduced between the first and second operations to reflect the reduction in target pest numbers, dropping from 2 to 1.5 kg/ha for pre-feed, and from 4 to 2 kg/ha for toxic. As part of the 1080-to-zero protocol, the bait lure was also changed to reduce bait-shyness. The first operation used cinnamon-flavoured 6 g RS5 cereal pellets, while the second used orange lured 6 g Whanganui #7 cereal pellets. Helicopters used a 50% flight path overlap for both pre-feed and toxic applications to provide an even distribution of bait and reduce gaps (Figure 2).

In the same year, as part of a 3-yearly pest control plan, the rest of the national park received a standard 1080 application [24] following the Department of Conservation’s protocols [29]. Non-toxic pre-feed was applied once on 18–19 July and toxic bait on 26–27 June. Both the pre-feed and toxin applications used cinnamon-flavoured RS5 6 g cereal pellets with a sow rate of 2 kg/ha. The helicopter paths had no overlap.

For both the 1080-to-zero and standard applications, the bait had a lure concentration of 0.3%, and toxic pellets had a 1080 concentration of 0.15%. Bait was supplied by Orillion from Whanganui, New Zealand, and manufactured between 2.3 and 5.6 months before application. Bait was only distributed in fine weather, and all operations had dry weather in the week following each application.

### 2.3. Pest-Monitoring Timeline

Monitoring of black rats (*Rattus rattus*) was repeated 10 times between February 2019 and February 2021: twice prior to the 1080 applications, 3 weeks after each toxic application and every 3 months thereafter until February 2021. This provided pre-1080, immediately post-1080 and approximately 15 months of recovering pest densities. Although the rate of toxin breakdown is variable depending on environmental conditions [30], we selected a 3-week monitoring stand-down period after each toxic application to reduce the potential for ongoing toxin exposure and to minimise the potential for device avoidance [31,32] due to acute stress from the operation. Monitoring after the standard operation was slightly delayed due to logistical complications, taking place 4.8 weeks after the toxic application compared to 3.3 and 2.7 weeks for the two 1080-to-zero operations. For each of the 10 trips, monitoring was carried out at both study sites, except for May 2019, when flooding prevented access to the standard treatment site. For logistical reasons, both sites could not be observed at once, and measurements for each site were on average 13 and no more than 20 days apart.

### 2.4. Rat Mark-Recapture

Rat density was measured using mark-recapture over 5 consecutive nights, except for one instance where trapping was cut short at 4 nights, and another instance where trapping was suspended for 2 nights, both at the standard treatment site due to heavy rain. Live capture Tomahawk model 201 traps were spaced 25 m apart in a 7 × 7 grid (*n* = 49 traps), baited with a 50:50 Pic’s^TM^ peanut butter and oat mixture. For the first 1080-to-zero monitoring session, the traps were covered with vegetation for rain shelter, and hay bedding was provided for warmth. However, the rats frequently pulled the vegetation inside the trap, and the hay became cold when wet. Despite trapping in late summer, this was not sufficient to prevent cold exposure. For all trapping thereafter, a white corflute rain cover was fitted over the back of the traps, and the hay bedding was replaced with water-resistant Dacron^TM^ stuffing. With these measures, evidence of cold exposure was rare even in winter. Traps were checked every morning as per the conditions of our animal ethics permissions (University of Auckland Animal Ethics Committee Approval No. 8442).

Each captured rat was identified according to its species and transferred to a plastic bag where it was weighed using a spring scale and then anaesthetised to effect using hand-pumped isoflurane gas [33]. While unconscious, its sex was identified, and females were checked for fur wear around the nipples—an indicator of suckling pups and thus breeding. While still unconscious, the animal was then fitted with a uniquely numbered metal ear tag for individual identification and returned to the cage to recover. The anaesthesia provided pain relief during ear tagging and had minor memory-loss effects, which improve the recapture rate [34]. Upon full recovery from the anaesthesia, the animal was released at the site of capture. For each recapture, the trap number and ear tag ID were recorded.

### 2.5. Relative Abundance Indices

Relative abundance indices (i.e., percent of devices detecting rats) were obtained using two different devices: wax tags and tracking tunnels. Wax tags were used to supplement the rat mark-recapture monitoring [35]. On the first night of each trapping session, unscented wax tags were attached to a tree or tree fern, 30 cm off the ground, within 1 m of each rat mark-recapture trap (*n* = 49 tags). Wax tags were recovered after 7 nights, except for two instances at the standard treatment site where flooding disrupted site access, in which case they were recovered after 5 nights. Wax tags were assessed for rat chew. Rat chew was differentiated from mouse chew by having an incisor width of 0.6 mm or more, determined using a 0.6 mm wire for a reference in the field [36].

From November 2019 onward, tracking tunnels were added to the study to improve the detection of rats after the 1080 treatment [37]. Tracking tunnels were thus present for six of the ten monitoring events at each site. At each site, we used the closest existing Department of Conservation tracking tunnel line (one line per site, 297 metres from the standard treatment site and 235 metres from the 1080-to-zero site). Each line was approximately 1 km in from the forest edge and had 10 tunnels spaced 50 m apart. Cards were deployed for one dry night during mark-recapture trapping. Pre-inked Black Trakka^TM^ cards were installed baited with peanut butter in the centre of the card and retrieved the next day.

### 2.6. Mark-Recapture Statistical Analysis

Mark-recapture data were analysed using the package ‘secr’ in R software to model spatially explicit capture-recapture with half-normal detection curves and full-likelihood [38]. Captures of non-target species were excluded from analysis. Covariates across mark-recapture sessions were applied to density, g0 (capture probability) and *σ* (scale of movement) to estimate pre-1080 densities, the 1080 impact and the recovery thereafter at each site.

The covariates of interest were: 1080 effect (0 for sessions prior to the first 1080 application and 1 thereafter), linear recovery (days since the most recent 1080 application), accelerating recovery (the square of days since 1080), diminishing recovery (the square root of days since 1080), season, mark-recapture trip (numbered 1 to 10) and site (1080-to-zero vs. standard). Accelerating and diminishing recovery were always applied in additive combination with linear recovery. The effect of learnt behaviour ‘b’ was also investigated for g0.

Due to the large number of covariates, it was not possible to model every covariate combination. Instead, six sequential model sets were used to target different aspects of the model. The difference in the Akaike Information Criterion adjusted for small samples (dAICc) was used to compare models. At each stage, models with a dAICc > 10 were removed from further analysis to avoid excessive model proliferation.

Density was investigated first. Holding g0 and *σ* constant, all additive covariate combinations for density were modelled (model set 1a, 51 models). A subset of these models, only considering temporal covariates, was also assembled (model set 1b, 27 models). This was followed by a preliminary investigation of g0 and *σ*. Density was modelled using the best-ranked covariate combination from model set 1a, and each covariate was added to g0 (model set 2a, 8 models) or *σ* (model set 2b, 7 models) while holding the other constant. At this stage, site was identified as a highly ranked covariate, with the potential to affect all of density, g0 and *σ* (see results). For each of the most supported models (dAICc < 10) from sets 1a and 1b, every combination of site across density, g0 and *σ* was generated (model set 3, 40 models).

The most supported models from model set 3 became the ‘base models’ for further refinement of g0 and *σ* (model set 4, 104 models). For each base model, each covariate was added independently to g0 or *σ*. Any addition which was supported (AICc less than or within 2 of the base model) was noted, and all additive combinations thereof were generated. All models from this set were then compared to identify the most supported covariate combinations overall.

From there, interactions with site were explored using the most supported models from model set 4 (model set 5, 54 models). Where site and another covariate influenced the same variable, the interaction between them was modelled. Where there were multiple possible interactions with site, each was modelled pair-wise independently. Any interactions which improved the model (AICc within 2 of its exclusion) were then applied together as multiple pair-wise interactions with site. All models from this set were then compared.

Finally, a new covariate was generated: ‘method’, with a value of 1 for the first 1080-to-zero session and 0 for all others. This reflects the slightly different trap set-up used in that monitoring session. The ‘method’ covariate was added to all of the most supported models from set 5 (model set 6, 28 models). The final model with the lowest AICc score was selected from this group.

### 2.7. Relative Abundance Indices Statistical Analysis

The relative abundance index (i.e., percent of devices detecting rats) was calculated separately for wax tags and tracking tunnels, and for each monitoring session at each site. The results for each site were graphed for visual comparison with density estimates. The results from the two sites were then pooled for correlation testing. The Pearson correlation test in R software was used to assess the linear correlation between wax tag index and density estimates, tracking tunnel index and density estimates, and wax tag and tracking tunnel indices [39]. The relationship between the density estimates and tracking tunnel and wax tag indices was then visualised using a fitted least-squares linear regression line drawn with the package ggplot2 in R software [40].

## 3. Results

### 3.1. Mark-Recapture Trapping

All rats captured were identified as black rats (*Rattus rattus*). There was no apparent sex bias. Excluding sessions with fewer than 10 individuals, females composed one to two thirds of captures per session, and overall half of all captures in the study. Based on the presence of nipple fur wear for females and descended tests for males, 70 g represented the lowest recorded weight for sexual maturity, but reproduction was not common below 120 g. Breeding occurred primarily in summer and autumn, with fur wear around females’ nipples twice as common during these seasons than in winter and spring. Smaller juveniles (30 to 70 g) representing recent breeding composed 15% of captures in summer and 16% in autumn, but only 3% in winter and 6% in spring. Larger juveniles (70 to 120 g) increased from 14% to 26% of captures from summer to autumn, then decreased to 20% in winter and 0% in spring. These demographic trends did not differ notably between years or sites.

The 1080-to-zero and standard treatment sites, respectively, recorded 544 and 216 rat captures of 203 and 126 unique individuals. Monitoring immediately after 1080 returned one, two and zero rats for the two 1080-to-zero applications and one standard application, respectively. No individuals captured before the standard application or first 1080-to-zero application were recaptured afterwards. However, one individual captured between the two 1080-to-zero applications was recaptured after the second application. In total, there were three sessions with zero rats captured, all at the standard treatment site after 1080.

### 3.2. Relative Abundance Indices

The Pearson linear correlation tests returned high correlation coefficients (r) between density estimates and relative abundance indices (Figure 3). Rat density and tracking tunnel index were more strongly correlated (r = 0.92, *p* < 0.001) than rat density and wax tag index (r = 0.64, *p* < 0.01), but both were significant. Tracking tunnels and wax tags also showed a significant correlation (r = 0.82, *p* < 0.01). There was evidence of differential performance of relative abundance indices at very low or very high densities. At low density, on three occasions, wax tags returned no rat sign when rats had been identified via trapping or tunnels. On one occasion, tunnels returned rat sign when rats had not been identified via trapping. At high density, tracking tunnels were saturated, returning 90% or 100% rat detection. Wax tags did not saturate but may lose linearity at very high densities, although this is based on only a small number of such observations (*n* = 2).

### 3.3. Density Estimates

In total, 242 models were run across 6 model sets (the exhaustive list is in the Appendix A). During early analysis, the accelerating recovery covariate returned exceptionally poor results (dAICc > 650) when applied to density, g0 or σ alone or with another temporal covariate. This covariate was thus removed from the analysis.

The investigation of density while holding g0 and σ constant returned two competitive models when including site (model set 1a) and three when excluding site (1b) (Table 1). These tended to vary density by 1080, diminishing recovery and site. Where site was excluded (set 1b), it was replaced by trapping or season. One model excluded the 1080 covariate and one supported a linear recovery.

The exploration of g0 and σ using the most supported model from set 1a revealed a strong effect of site on g0 and σ (model sets 2a and 2b, Table 2). When compared, the effect of site on g0 was stronger than σ (dAICc 32.58). Further exploration of site across density, g0 and σ used all four density configurations identified in set 1a and 1b (model set 3, Table 2). Six competitive models were identified, all of which included site on g0, and some of which also included site on density, σ or both. These included five density configurations: all shared the diminishing recovery scenario, most included the 1080 suppression effect, and some included an effect of site, trip or both.

Model set 4 added further covariates to the competitive models from set 3 (Table 3). The resulting models ranked quite closely, with 17 modes with dAICc less than 10 and 8 with less than 5. The model configuration varied: the only universal features were an effect of site on g0 and diminishing recovery on density. The addition of trip, linear recovery or b often ranked well for g0, and sometimes trip or 1080 for *σ*.

When investigating interactions with the site covariate, most competitive models returned an interaction with 1080 and/or linear recovery for density (model set 5, Table 4). There was no difference between including a site interaction for 1080 and linear recovery (dAICc 0), or just for 1080 (dAICc 0.07). This continued into the final model set, where three models returned a dAICc of less than two, and the most supported of these failed to achieve a model weight (AICwt) of more than 0.5. Of these three models, the two most supported included an interaction between site and 1080, and the third also included an interaction with linear recovery. All three models supported the addition of method to g0. The highest ranked model was selected for graphing and analysis.

Throughout the model building process, a diminishing recovery rate of density was consistently supported. A site effect on g0 was also supported, with the standard treatment site having a lower capture probability than the 1080-to-zero site. The movement parameter *σ* tended not to be influenced by covariates, and by the final model set all of the most supported models (dAICc < 10) held *σ* constant. Based on the most competitive (dAICc < 2) models from the final model set, there was strong support for an interaction between site and 1080 on density. There was weak support for an interaction between site and linear recovery, present in one of the three models. There was moderate support for a linear and learnt behaviour effect on g0, or a trip effect. However, the final g0 covariate configuration had little effect on g0 values.

In the final most supported model, *σ* was constant, at a value of 22.24 (20.63–23.98, 95% lcl to ucl). The standard treatment site had a much lower g0 than the 1080-to-zero site, with a capture probability ranging between 0.03 and 0.05 versus 0.09 and 0.16 across all ten sessions. Capture probability increased over time at both sites, regardless of 1080 (Figure 4). The model found an effect of method on g0: the different capture method used in the first monitoring session at the 1080-to-zero site returned a much higher capture probability than the second (0.16 vs. 0.09), but with g0 increasing over time the final capture probability was similar to the first (0.15).

Density was higher at the standard site than the 1080-to-zero site before 1080 (25.84 vs. 14.15 rats/ha) but similar 3 weeks after treatment (0.51 vs. 0.72 and 0.77 rats/ha for the standard and two 1080-to-zero applications) (Figure 5). Using the density estimates and effective sampling area (4.35 ha), an estimated 62 and 112 individuals were present at the 1080-to-zero and standard treatment site before 1080 application. Based on these estimates, 65% and 38% of the resident 1080-to-zero and standard site populations were captured in the final mark-recapture session prior to treatment. The 1080 treatment resulted in an estimated 95% and 98% reduction in population at the 1080-to-zero (first application) and standard site. At both sites, population growth occurred rapidly between January and August 2020 and then plateaued, 9 months after the second 1080-to-zero application, and 14 months after the standard application. At the 1080-to-zero site, a slight density increase was seen in the last capture session, but it is unknown if this indicates further population growth after the monitoring ended. The standard treatment site may have been slightly slower to recover, but large error bars preclude a fine-scale comparison, and the final model did not support an interaction between site and recovery rate. This indicates that any site difference in recovery was minimal. The highest densities recorded following 1080 were slightly lower at the standard site (11.78 vs. 13.04 rats/ha, standard vs. 1080-to-zero), although the variation was high and the error bars overlap. These represent a return to 92% and 43% of pre-1080 densities at the 1080-to-zero and standard sites. By the end of the study, although final densities at each site were similar, only the 1080-to-zero site had returned to densities similar to those recorded before 1080, while the standard treatment site had not.

## 4. Discussion

The assessment of invasive species management tools ideally requires a clear measurement of pest densities immediately before and after treatment (‘short term’), and the rate of recovery thereafter (‘long term’). Without this information, it is not possible to determine the magnitude or duration of effect the tool had. We used extended rat monitoring to compare the short- and long-term efficacy of a novel 1080-to-zero aerial toxin application to a standard 1080 application, in the context of black rat (*Rattus rattus*) management.

Rat density can be influenced by a number of factors, including habitat [41], predation [42] and food abundance [10]. Densities also fluctuate seasonally and yearly [42]. The maximum densities observed both before and after 1080 were generally consistent with the upper estimates previously recorded for North Island’s mixed and podocarp forests (0.29 to 13.61 rats/ha [42]; 3 to 12 rats/ha [43]). However, the pre-1080 density at the standard treatment site was amongst the highest ever recorded on the New Zealand mainland (17 rats/ha, in preparation [44]; 22 rats/ha [45]). The reason for this is unclear, but may reflect the 2019 mast year [27]. At both sites, fruit was observed in greater abundance in 2019 than 2020 or 2021, with the standard treatment site displaying a greater diversity of fruit than the 1080-to-zero treatment site. Masting has been observed to elevate rat abundance in beech and southern rimu forests [10,44,46], although the effect in North Island mixed podocarp forests is less clear [42,47].

Both the 1080-to-zero and standard 1080 treatments achieved a >95% reduction in rat densities and a reduction of wax tag detections to zero, on par with common benchmarks for success (>80% [48] or >90% reduction in abundance [29]; <5% post-treatment relative abundance indices [10]). Rat density then recovered rapidly before plateauing, suggesting a stabilisation of the populations and a final recovery time of 9 and 14 months since the 1080-to-zero and standard applications, respectively. This recovery time is within the mid-range observed for this habitat type, with previous studies reporting rat populations recovering within 4 months [49] to 2.5 years [47]. Interestingly, according to the final density model, rat movement patterns did not change with the application of 1080 or across the wide range of densities observed. In other mammal species, home range is known to change with density [50]. Black rats are known to display varying home ranges between sites, although it is unclear to what extent this is an influence of density or food abundance [51,52]. However, in this study there may have been too few captures at low densities to reliably detect a change in movement activities.

A combination of live traps, tracking tunnels and wax tags was used to improve the detection of survivors. This redundancy proved useful, with tracking tunnels detecting individuals when traps and wax tags did not. The apparent sensitivity of tracking tunnels contrasted with concerns from others, who observed that some rats will avoid entering them [31]. Despite the high sensitivity observed here, it is likely there will always be some individuals that avoid devices and will only be detected by devices that do not require interaction (e.g., cameras) [32,33]. Interestingly, the modifications to traps between the first and second sessions at the 1080-to-zero site showed strong effects on capture probability. Covering traps with brush may reduce neophobic avoidance and be a preferred trapping method when monitoring low-density or trap-avoidant rat populations [53] (but see [54,55]). Overall, the relative abundance indices obtained in this study appeared highly correlated to density. However, there was evidence that this relationship may wane at very high densities through the saturation of devices. The wax tags also tended to return false absences. This is consistent with other studies that show strong correlation of relative abundance indices with density and trapping rates at moderate densities [35,37], but poor correlation at very high or low densities [56,57].

Overall, monitoring indicated that the 1080-to-zero applications did not achieve rat elimination, with at least one rat captured 3 weeks after each 1080-to-zero application. However, by itself, this does not necessarily indicate a failure to meet the objectives. A secondary goal of the 1080-to-zero application, where it fails to eliminate pests, is to reduce them to near-zero levels that can potentially be eradicated, afterwards, with more targeted methods, such as spot toxin treatment [21,58]. However, the results obtained here cannot be construed as near-zero, with post-treatment densities of approximately 0.7 rats/ha translating to approximately 1540 rats across the 2200 ha Kaitake Ranges. The second 1080-to-zero application did not improve circumstances, with at least one confirmed survivor and similar post-treatment densities to the first application. Ultimately, the 1080-to-zero treatment did not appear to improve rat suppression relative to the standard method. Post-1080 rat densities were similar at the two sites and the final model did not support a difference in recovery rate. One out of the three most supported models suggested a slower recovery at the standard treatment site, but it seems unlikely that this difference in recovery is meaningful. Even if it were, it would indicate an improved outcome at the standard treatment site.

1080 operations can be influenced by local factors [10], and the possibility that site differences may have masked an improved impact from the 1080-to-zero operation must be considered. The only differences highlighted by the final density model were a higher pre-treatment density and lower capture probability at the standard site than the 1080-to-zero site. High pest densities are sometimes associated with worse outcomes for 1080 operations [10,48], and a low capture probability could indicate neophobic behaviour that translates to reduced bait consumption [55,59]. However, if these factors were influencing the 1080 outcome, this should have only worsened the outcome at the standard site, and so they are unlikely to have influenced our conclusions. Thus, although clear differences among the sites existed, for the parameters we measured they should not have predisposed the 1080-to-zero site to a higher failure rate. However, that is not to say that some other site-specific factors may not have contributed to the unsuccessful outcome of the 1080-to-zero application.

A key question in pest elimination monitoring is whether the individuals observed soon after treatment are survivors or invaders. With no barrier to movement, the primary protection against reinvasion was the distance from the study site to the treatment boundary (837 m for the 1080-to-zero site). The invasion behaviour of black rats is still not well understood. While some studies indicate that movement over this scale could take weeks to months [9,24,47], others suggest it could take days [60]. Reinvasion could not be ruled out in this study, as no rats captured prior to the first 1080-to-zero application were recaptured later. However, only 65% of the population was captured before treatment. Additionally, at least one individual survived the second 1080-to-zero application, having been previously captured after the first application. Furthermore, rats are known to display varying responses to management devices, and rats which do not enter traps may also be pre-disposed to avoid toxic bait and so survive 1080 applications [61]. Future studies may benefit from supplementing mark-recapture monitoring with non-invasive detection devices (e.g., trail cameras) in the first few weeks immediately after treatment [54,62].

The results seen here differ from the only other large-scale 1080-to-zero operation completed so far: a South Island operation which did not achieve full removal of rats, but did reduce them to near-zero levels [21]. That operation also saw improved rat removal from a second toxin application [21], something echoed in prior studies which explored double-1080 applications using standard application methods [9]. It is not unheard of for similar 1080 operations to produce variable outcomes from site to site for no apparent reason [10]. The results observed here highlight the complexity of aerial toxin applications, and the need for repeated testing to assess the variability in performance across a range of environments.

Without replication, our study cannot generalise which factors may influence elimination success for 1080-to-zero operations, but can point to key features that should be considered in future research. The study sites showed features associated with difficult-to-manage locations. This includes high food abundance [63], high pest density and rugged terrain [10]. The influence of these factors on 1080 operations is poorly understood, although they are associated with a higher risk of failure [10,63]. Additionally, the timing of the operations was not optimal. The first 1080-to-zero application was performed in autumn following a large mast fruiting event, with fruit visible on the ground. The high food abundance [63] and the application of bait in summer or autumn instead of winter or spring can reduce bait uptake [64]. The second 1080-to-zero application also had long delays between the two pre-feeds (82 days) and pre-feed and toxin (43 days). These were longer than recommended (5 to 14 days in mast years [29]) or used in other operations (e.g., 5 to 10 days [22]; 26 days [24]; 15 to 28 days [21]). Pre-feeding is critical in a double toxin application as a mechanism for reversing bait shyness in survivors of previous treatments [9]. Whether such a delay would have impacted the efficacy of the pre-feed is unclear. Future studies should continue to investigate the influence of factors such as habitat, pest density, food abundance and operational timing on 1080-to-zero and other eradication operations [65].

Finally, further modification of the 1080-to-zero method may improve results. In particular, while bait switching between the first and second application is a standard method for reducing bait shyness [21], Nugent et al. recently found that this is most effective if individuals are exposed to both baits before being exposed to toxin [66]. Operations may thus benefit from pre-feeding with both bait types before any toxin is applied. Additionally, there is evidence that targeted bait deployment can improve bait distribution and pest reduction [48]. Although targeted bait deployment was initially explored as a method for reducing bait sow rate while maintaining adequate rat suppression, improved results may be seen from combining it with the increased sow rate of the 1080-to-zero method. This may be especially relevant for rugged terrain, such as that of the Kaitake 1080-to-zero treatment area, where the uneven topography may inhibit even bait distribution [10].

## 5. Conclusions

This study assessed the short- and long-term outcomes on rat density of a novel eradication 1080-to-zero application technique, in comparison to a standard 1080 application. In these Taranaki study sites, the novel method did not appear to achieve elimination of rats nor improve results relative to the standard method. However, with only one 1080-to-zero treatment site, this study cannot make generalised statements regarding the efficacy of this tool. Operations using 1080 can produce variable outcomes, often for unknown reasons, and the results here differ from those in a similar South Island operation. Future studies should investigate how habitat and operational design influence the efficacy of this and other eradication tools.

## Figures and Tables

**Figure 1 animals-12-00309-f001:**
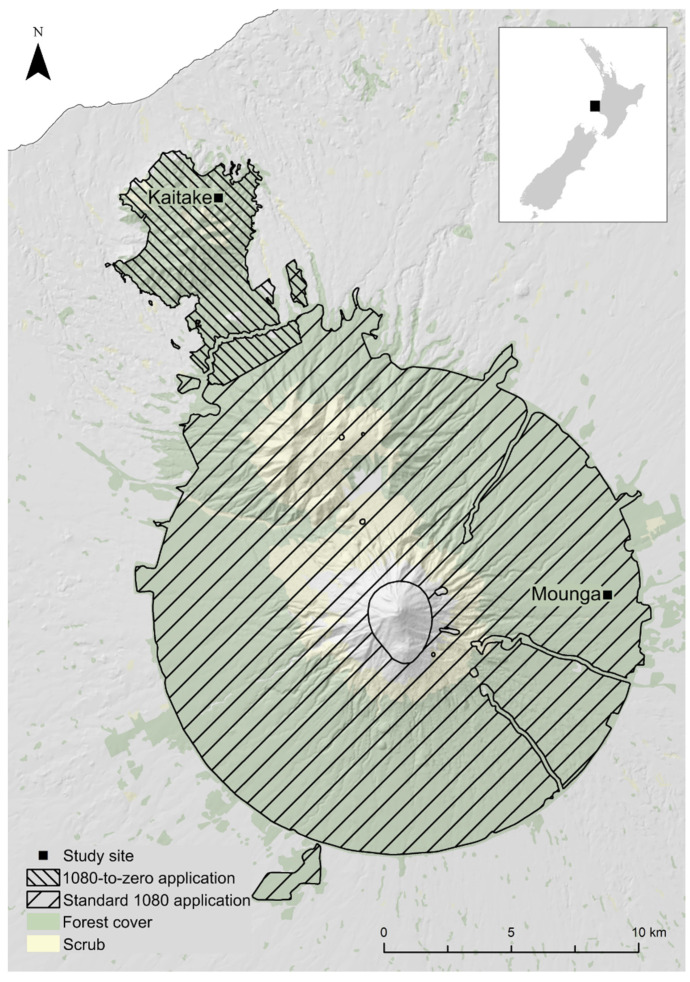
1080 treatment areas in the national park Te Pakakura o Taranaki. The lines indicate the treatment area for two aerial 1080 application methods: a novel 1080-to-zero method and a standard method. The total 1080 treatment area approximates the boundaries of the national park. Rat monitoring sites (Kaitake and Mounga) are indicated by black boxes.

**Figure 2 animals-12-00309-f002:**
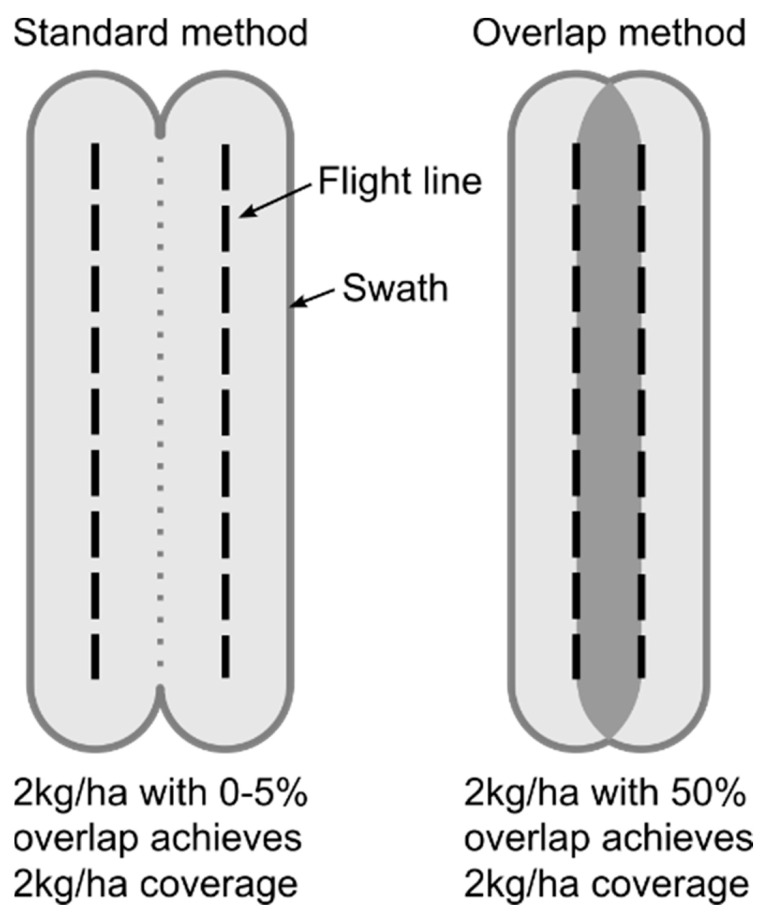
Helicopter flight paths for the 1080 bait applications. Standard 0% overlap flight path (**left**) versus 1080-to-zero 50% overlap (**right**).

**Figure 3 animals-12-00309-f003:**
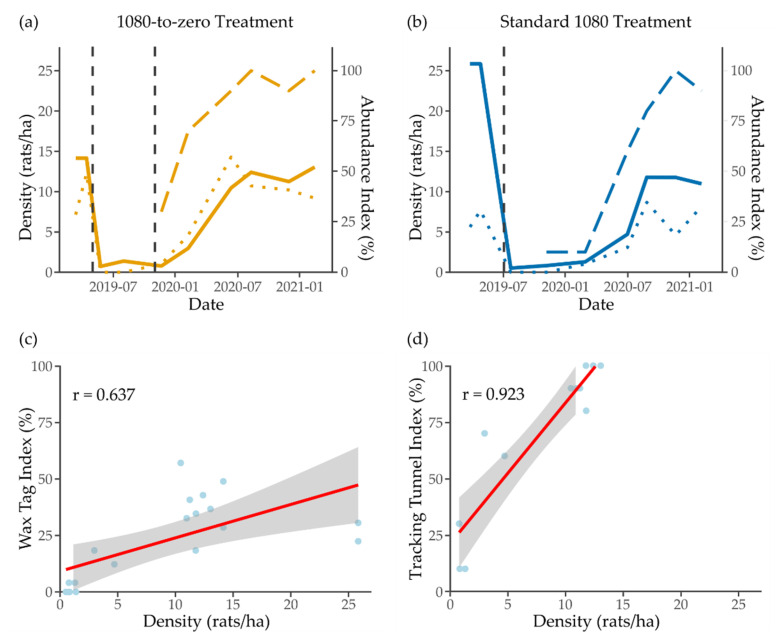
Density estimate (solid line), wax tag index (dotted line) and tracking tunnel index (long dashed line) for the (**a**) 1080-to-zero site and (**b**) standard site. Fitted least-squares linear regression (red) with 95% confidence interval (grey) and Pearson coefficient of correlation (r) between density estimates and (**c**) wax tag index and (**d**) tracking tunnel index.

**Figure 4 animals-12-00309-f004:**
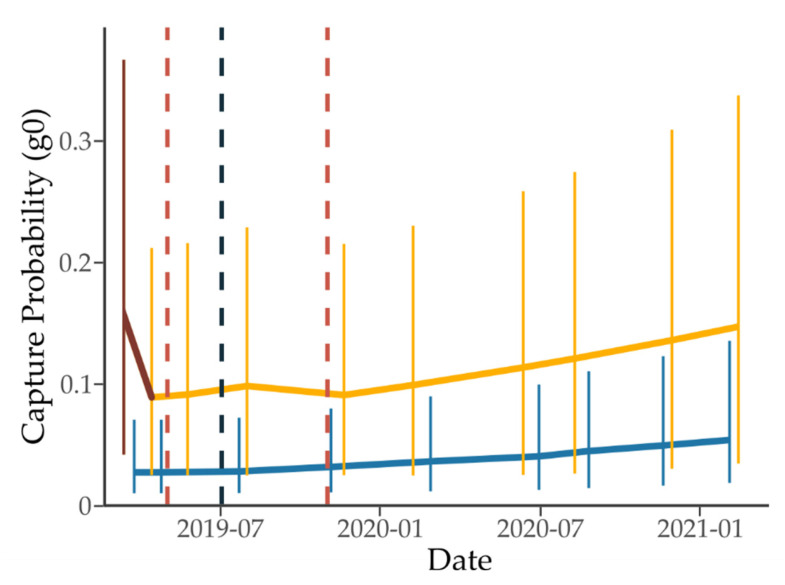
Capture probability (g0) estimates at the 1080-to-zero (lighter orange) and standard treatment (darker blue) sites. Error bars indicate 95% lower and upper confidence intervals. The darker brown colour indicates the different trapping method used for the first 1080-to-zero monitoring session. Vertical dashed lines indicate the toxic application at each site (lighter red for 1080-to-zero, darker blue for standard).

**Figure 5 animals-12-00309-f005:**
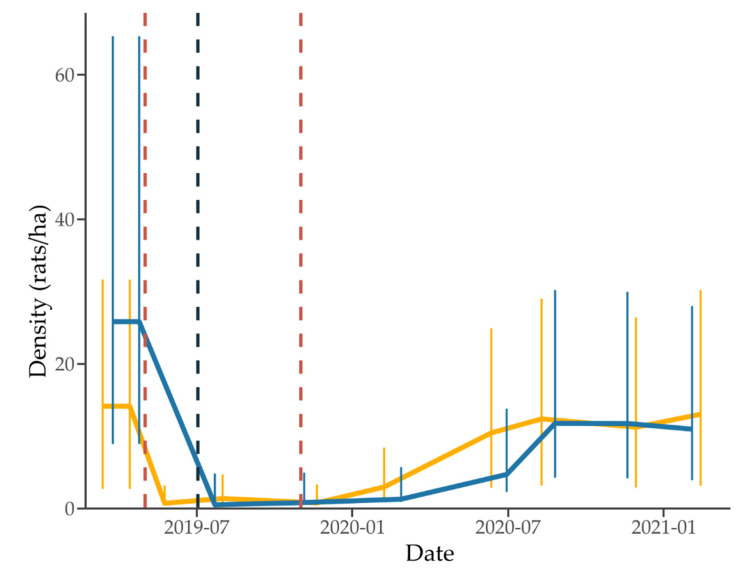
Rat mark-recapture density estimates at the 1080-to-zero (lighter orange) and standard treatment (darker blue) sites. Error bars indicate 95% lower and upper confidence limits. Vertical dashed lines indicate the toxic application at each site (lighter red for 1080-to-zero and darker blue for standard).

**Table 1 animals-12-00309-t001:** Summary of model set 1 (all models with dAICc < 10).

**Set 1a: Exploring density using all covariates**			**dAICc**	**AICcwt**
D~ 1080 + days + sqrt(days) + site			0	0.80
D~ days + sqrt(days) + site			2.77	0.20
**Set 1b: Exploring density while excluding site**			**dAICc**	**AICcwt**
D~ 1080 + days + sqrt(days) + trip			0	0.73
D~ 1080 + days + season + trip			2.14	0.25
D~ 1080 + season + trip			6.78	0.02

**Table 2 animals-12-00309-t002:** Summary of model sets 2 and 3 (all models with dAICc < 10).

**Set 2a: Preliminary exploration of g0**			**dAICc**	**AICcwt**
D~ 1080 + days + sqrt(days) + site	g0~ site		0	1.00
**Set 2b: Preliminary exploration of *σ***			**dAICc**	**AICcwt**
D~ 1080 + days + sqrt(days) + site		*σ*~ site	0	0.95
D~ 1080 + days + sqrt(days) + site		*σ*~ days + sqrt(days)	6.45	0.04
D~ 1080 + days + sqrt(days) + site		*σ*~ post1080	8.85	0.01
**Set 3: Exploring site across density, g0 and *σ***			**dAICc**	**AICcwt**
D~ days + sqrt(days)	g0~ site		0	0.58
D~ 1080 + days + sqrt(days)	g0~ site	*σ*~ site	2.56	0.16
D~ 1080 + days + sqrt(days)	g0~ site		2.58	0.16
D~ 1080 + days + sqrt(days) + site	g0~ site		5.37	0.04
D~ 1080 + days + sqrt(days) + trip + site	g0~ site	*σ*~ site	5.78	0.03
D~ 1080 + days + sqrt(days) + trip	g0~ site		6.61	0.02

**Table 3 animals-12-00309-t003:** Summary of model set 4 (all models with dAICc < 10).

Set 4: Covariate combinations for g0 and *σ*			dAICc	AICcwt
D~ days + sqrt(days)	g0~ site + trip		0	0.38
D~ 1080 + days + sqrt(days)	g0~ site + days + b		1.17	0.21
D~ 1080 + days + sqrt(days) + trip	g0~ site	*σ*~ 1080	2.42	0.11
D~ 1080 + days + sqrt(days)	g0~ site + b	*σ*~ trip	4.20	0.05
D~ days + sqrt(days)	g0~ site + days		4.31	0.04
D~ 1080 + days + sqrt(days) + site	g0~ site + days + b		4.36	0.04
D~ 1080 + days + sqrt(days)	g0~ site + days		4.41	0.04
D~ 1080 + days + sqrt(days)	g0~ site	*σ*~ trip	4.55	0.04
D~ 1080 + days + sqrt(days)	g0~ site + days + b	*σ*~ trip	5.94	0.02
D~ days + sqrt(days)	g0~ site		6.75	0.01
D~ 1080 + days + sqrt(days) + trip	g0~ site	*σ*~ trip	7.01	0.01
D~ 1080 + days + sqrt(days) + site	g0~ site + trip		7.03	0.01
D~ 1080 + days + sqrt(days) + site	g0~ site + days		7.79	0.01
D~ 1080 + days + sqrt(days) + site	g0~ site + b		8.78	0
D~ 1080 + days + sqrt(days)	g0~ site + 1080		8.96	0
D~ 1080 + days + sqrt(days)	g0~ site	*σ*~ site	9.31	0
D~ 1080 + days + sqrt(days)	g0~ site		9.33	0

**Table 4 animals-12-00309-t004:** Summary of model Set 5 and 6 (all models with dAICc < 10).

**Set 5: Location interactions**			**dAICc**	**AICcwt**
D~ 1080*site + days*site + sqrt(days)	g0~ site + days		0	0.33
D~ 1080*site + days + sqrt(days)	g0~ site + days + b		0.07	0.32
D~ 1080*site + days*site + sqrt(days)	g0~ site + days + b		2.23	0.11
D~ 1080*site + days + sqrt(days)	g0~ site + days		3.56	0.06
D~ 1080 + days*site + sqrt(days)	g0~ site + days		3.85	0.05
D~ 1080 + days*site + sqrt(days)	g0~ site + days + b		4.01	0.04
D~ 1080*site + linear*site + sqrt(days)	g0~ site + b		4.33	0.04
D~ 1080*site + days + sqrt(days)	g0~ site + b		5.70	0.02
D~ 1080*site + days + sqrt(days)	g0~ site + trip		6.85	0.01
D~ days + sqrt(days)	g0~ site + trip		7.05	0.01
D~ 1080 + days*site + sqrt(days)	g0~ site + b		8.02	0.01
D~ 1080 + days + sqrt(days)	g0~ site + days + b		8.22	0.01
D~ 1080*site + days*site + sqrt(days)	g0~ site + trip		8.58	0
D~ 1080 + days + sqrt(days) + trip	g0~ site	*σ*~ 1080	9.47	0
**Set 6: Incorporating method**			**dAICc**	**AICcwt**
D~ 1080*site + days + sqrt(days)	g0~ site + days + b + method		0	0.41
D~ 1080*site + days + sqrt(days)	g0~ site + trip + method		0.64	0.30
D~ 1080*site + days*site + sqrt(days)	g0~ site + days + b + method		1.19	0.23
D~ 1080 + days*site + sqrt(days)	g0~ site + days + b + method		5.51	0.03
D~ 1080*site + days + sqrt(days)	g0~ site + days + method		7.20	0.01
D~ 1080*site + days*site + sqrt(days)	g0~ site + trip + method		7.59	0.01
D~ 1080*site + days*site + sqrt(days)	g0~ site + days		8.48	0.01
D~ 1080*site + days + sqrt(days)	g0~ site + days + b		8.55	0.01

## Data Availability

Data are available in a publicly accessible repository: auckland.figshare.com/articles/dataset/Rat_Response_to_Two_Different_Aerial_Toxin_Treatments/17149160 (accessed on 13 January 2022).

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
