# Peer review of "Assessing Two Different Aerial Toxin Treatments for the Management of Invasive Rats"

_animals, 2022, doi:10.3390/ani12030309_

Round 1
Reviewer 1 Report
The paper untitled Assessing two different aerial toxin treatments for management of invasive rats (Rattus rattus) by O’Malley and colleagues investigates a field experimental analysis testing the efficacy of two control methods at short- and long-term of invasive black rat populations on large areas in mainland New-Zealand.
I have no major comments. The paper is very well written, provides very interesting results based on high quality and robust methods and analyses, which allows a very good discussion of hypotheses the authors very well exposed in the introduction.
I have only one general comment on this study:
Did the authors had the opportunity to record some basic biometric measures/observations on trapped/marked/released individuals – for example body mass, sex, and some basic criteria for female reproduction such as nipple size/morphology (to identify lactating females).
This could give some indication of the situation of the age and/or sex structure of the populations at the different stages of the control operations; when reading the paper – I was wondering if high densities recorded during the pre-treatment phases were related to population at a peak of abundance with a majority of young/subadults? Was the reproduction at this moment under arrest or not (probably arrested at such high density!)? Which was the age/sex structure of the populations at the moment they recovered? Wouldn’t have it been interesting to test body mass (and including large females potentially pregnant) in the modelling strategy – both on g0 or more importantly on sigma? I am conscient that even adding one more variable in the analyses may have posed overfitting problem in the modelling strategy. But at least could the authors add a paragraph in the discussion/conclusion about the importance to take parameters of the population biology of the target species into account when mounting control operation (varying age and/or sex related spatial or trophic behavior and also considering reproductive status of individuals?)
I have only one minor remark:
1/ Lines 167- 169: “A 3-week delay for monitoring after each toxic application was selected to ensure no more individuals were being exposed to the toxin, and to allow survivors to overcome acute stress from the operation so as to reduce device avoidance.” Is there any reference for this statement?
Reviewer 2 Report
Manuscript ID: animals-1528976
Assessing two different aerial toxin treatments for management of invasive rats (Rattus rattus) Authors: Tess D R O’Malley *, Margaret C Stanley, James C Russell
Review
This is very well written manuscript. Problems with invasive rodents (rats and mice) are crucial to NZ, with their unique fauna, and the goal of eradicating invasive mammals is really ambitious one. I have seen how they are working towards it with trapping lines and volunteer employment, therefore I was keen to see, how aerial treatments work. Perfect presentation. So, with no doubts manuscript deserves publication after minor revision.
Few remarks for changes to be done and some to be discussed.
- Could authors agree to eliminate (Rattus rattus) from the title? In fact, aerial treatment work for all three rat species, right?
- If yes, then Rattus rattus should be used as a first keyword.
- Now, on the name of this species. I know in NZ this is ship rat – however, internationally preferred common name is black rat (https://www.cabi.org/isc/datasheet/46831; https://australian.museum/learn/animals/mammals/black-rat/; https://www.itis.gov/servlet/SingleRpt/SingleRpt?search_topic=TSN&search_value=180362#null; https://www.mammal.org.uk/species-hub/full-species-hub/discover-mammals/species-black-rat/; https://www.gbif.org/species/2439270; https://www.ncbi.nlm.nih.gov/Taxonomy/Browser/wwwtax.cgi?mode=info&id=10117; https://newzealandecology.org/nzje/3261) though is also called house rat (https://www.iucnredlist.org/species/19360/192565917; https://animaldiversity.org/accounts/Rattus_rattus/ )
- Lines 47–48: please list these species one by one
- Line 54: could “million” be used?
- Line 93: abbreviation ZIP was not used in the text further
- Line 99: punctuation incorrect
- Line 108: after first use, species name could be used as rattus
- Line 117: space needed between 330 and m
- Line 141: remove words “personal communication”, as you cite it as source
- Chapter 2.5: please explain what is abundance index here, not in 2.7
- Line 223: maybe “squared number of days since”?
- Line 261: relative abundance, as you are speaking about an index
- Line 272: rattus
- Figure 3: I expect X and Y axis visible, both
- Figure 4: I expect X and Y axis visible
- Figure 5: I expect X and Y axis visible
- Line 399: authors personal observation is the part of manuscript, I think referring is not needed
- Line 503: Nugent et al.
- Back matter: I appreciate disclosing conflict of interests, but further standard phrase “funders had no influence on …” is expected
- Journal titles must be abbreviated in the References
- Line 555: https://doi.org/10.1016/S0006-3207(00)00188-9
- Line 567: long dash separating page numbers
- Please check, if issue number (in parentheses) should be given, or volume number only?
- Line 577: correct doi is https://dx.doi.org/10.20417/nzjecol.45.5
- Line 599: 1–7. https://doi.org/10.20417/nzjecol.44.13 instead of 1-7. https://doi.org/10.20417/NZJECOL.44.13
- Line 600: wrong punctuation „2020 – 2050“, original shows different
- Lines 602, 604, 608 etc,: why DOC. in the beginning of reference?
- Line 613: Rattus rattus
- Line 618: Rattus rattus, Mus musculus. Please check doi – result is J
- Line 628: Rattus rattus
- Line 635: (in prep.)
- Line 639: two mistypes in species names
- Line 645: Rattus rattus
- Line 648: Rattus rattus
- Line 650: is this doi correct? See https://doi.org/10.2307/24053665
- Line 654: Rattus rattus
- Line 655: is this doi correct? See https://doi.org/10.2307/26198754 journal says: DOI: 10.20417/nzjecol.40.25
- Line 657: Rattus rattus
- Line 661: Rattus rattus
- Line 664: DOI: https://doi.org/10.20417/nzjecol.42.3
- Line 665: Rattus rattus
- Line 684: Rattus rattus
- Line 685: correct doi is 2408. https://doi.org/10.1007/S10530-011-0051-6
- Line 690: 1–10.
- Line 694: journal says doi is https://doi.org/10.1016/j.biocon.2014.10.014
- Line 702: 1–6.
Reviewer 3 Report
This is a particularly interesting paper that tests the efficacy of two eradication methods using 1080. The authors have found that regarding Rattus rattus, the most elaborate 1080-to-zero aerial toxin application was not significantly different from the more standard 1080 application. This appears to question the efficacy of this methods but also calls for the interplay of several factors that may have influenced the results and need to be considered for further applications. One of the factors that appears to be important is rat density and abundance. This difference between the two sites where the two methods were applied may account for insignificant differences. Lower densities and abundances are always faster and easier to attend during population recovery than higher ones, other conditions being equal (which is rarely or never the case). The authors also tested the different methods of detection of individuals, as this is fundamental for attesting or not the efficacy of an eradication method. They found that tracking tunnels may provide the best estimates but at high densities all methods are proven inadequate. Supplementation by non-invasive methods may increase detection rates and provide more solid estimates. This needs to be proven in the field during next applications.
The authors admit that pest eradication, especially that of rats, is extremely difficult because of the biology of the species and lack of complete understanding of its biology of invasion. More standardized protocols involving and controlling for more factors need to be considered next time, always taking into account local factors that appear to be important for the success of the application. The paper deals adequately with these issues. Overall, it is an excellent paper that deserves publication.
Author Response
Thanks you very much for your revision! We appreciate the kind words and your support for publication of this paper.